# What do we really know about the appropriateness of radiation emitting imaging for low back pain in primary and emergency care? A systematic review and meta-analysis of medical record reviews

**Gabrielle S. Logan** [1]*, **Andrea Pike**[2], **Bethan Copsey** [3], **Patrick Parfrey**[1], **Holly Etchegary**[1], **Amanda Hall**[1,2]

**1** Faculty of Medicine, Memorial University, St. John's, NL, Canada, **2** Primary Healthcare Research Unit, Memorial University, St. John's, NL, Canada, **3** Nuffield Department of Orthopaedics, Rheumatology and Musculoskeletal Sciences, Centre for Statistics in Medicine, University of Oxford, Oxford, United Kingdom

* glogan@mun.ca

## Abstract

### Background

Since 2000, guidelines have been consistent in recommending when diagnostic imaging for low back pain should be obtained to ensure patient safety and reduce unnecessary tests. This systematic review and meta-analysis was conducted to determine the pooled proportion of CT and x-ray imaging of the lumbar spine that were considered appropriate in primary and emergency care.

### Methods

Pubmed, CINAHL, The Cochrane Database of Systematic Reviews and Embase were searched for synonyms of "low back pain", "guidelines", and "adherence" that were published after 2000. Titles, abstracts, and full texts were reviewed for inclusion with forward and backward tracking on included studies. Included studies had data extracted and synthesized. Risk of bias was performed on all studies, and GRADE was performed on included studies that provided data on CT and x-ray separately. A random effect, single proportion meta-analysis model was used.

### Results

Six studies were included in the descriptive synthesis, and 5 studies included in the meta-analysis. Five of the 6 studies assessed appropriateness of x-rays; two of the six studies assessed appropriateness of CTs. The pooled estimate for appropriateness of x-rays was 43% (95% CI: 30%, 56%) and the pooled estimate for appropriateness of CTs was 54% (95% CI: 51%, 58%). Studies did not report adequate information to fulfill the RECORD checklist (reporting guidelines for research using observational data). Risk of bias was high

**Data Availability Statement:** All relevant data are within the paper and its Supporting Information files.

**Funding:** GSL received two student awards to fund her Masters research, the Translational and Personalised Medicine Initiative Student Award (website: http://www.nlsupport.ca/getdoc/a3881a94-9098-48c5-930c-4d8a39c475f1/Student-Funding.aspx), and the Dean's Fellowship Award from the Faculty of Medicine, Memorial University. The funders had no role in study design, data collection and analysis, decision to publish, or preparation of the manuscript.

**Competing interests:** The authors have declared that no competing interests exist.

in 4 studies, moderate in one, and low in one. GRADE for x-ray appropriateness was low-quality and for CT appropriateness was very-low-quality.

## Conclusion

While this study determined a pooled proportion of appropriateness for both x-ray and CT imaging for low back pain, there is limited confidence in these numbers due to the down-grading of the evidence using GRADE. Further research on this topic is needed to inform our understanding of x-ray and CT appropriateness in order to improve healthcare systems and decrease patient harms.

## Introduction

Guidelines for the assessment and treatment of low back pain (LBP) have been in circulation since the 1980s with more than 11 countries publishing their own LBP clinical guidelines in the last two decades.[1] While most early versions of LBP guidelines did not recommend routine use of radiographic imaging for assessment of LBP, there were discrepancies about when to image (e.g., some guidelines provided specific criteria or timeframes for imaging and others did not). In the 1980s and 1990s, x-ray imaging was commonly recommended in the assessment of LBP persisting longer than four weeks[1] and Computed Tomography (CT) was often recommended in patients experiencing neurological deficits, including radicular symptoms. [2,3] For the last 25 years, there has been increased congruence among LBP guidelines regarding when and under what circumstances to use diagnostic imaging. Since 2000, the recommendations typically state that diagnostic imaging is warranted only when patients with LBP present with red flag symptoms that suggest the presence of one of four known specific spinal pathologies (severe cauda equina, infection, fracture, and cancer).[4,5] Guidelines have also been updated with respect to the potential direct and indirect patient harms of diagnostic imaging, particularly x-ray and CT, as well as their lack of clinical utility for non-specific LBP. While MRI is another form of diagnostic imaging, it does not expose patients to the ionising radiation that x-ray and CT both emit; thus we are focusing only on those two imaging modalities.

### Harms of over-testing

**Patient harms.** Both x-ray and CT imaging expose patients to ionizing radiation, a known mutagen that can increase risk of cancer, with CT exposing patients to more radiation than x-ray.[6] The human body can tolerate some radiation, but the more exposure that a patient has to radiation, the greater their cancer risk. This risk of radiation is even greater to children and young adults as radiation can effect both male and female fertility.[7] Thus, radiologists typically recommend using x-ray and CT only when medically necessary and clinically justified to patient care.[8,9]

In addition to the harms from radiation, imaging can reveal incidental findings, such as anatomical abnormalities, that are extremely common in asymptomatic patients, and only weakly correlated with patient symptoms.[10] For example, a systematic review in 2014 found that disc degeneration was present in 96% of asymptomatic adults aged 80 and up, and disc bulges found in 80%.[11] Moreover, patients who receive diagnostic imaging do not have better patient outcomes compared to those treated without imaging.[5,10] Chou et al. performed a systematic review and meta-analysis to compare physical outcomes of patients with LBP who

received imaging to those who did not.[12] They found that patients who received immediate imaging for non-serious LBP had similar pain and function outcomes both in the short and long term compared to patients who received usual care without imaging.[12] The harm of incidental findings is that patients may have to be sent for further tests or procedures to confirm that the finding is in fact benign, which may delay the patient receiving the appropriate treatment.

**Health system burden.**   In addition to patient harms, over-testing results in a substantial economic burden to healthcare systems.[13] In the United States, the amount of dollars spent on all CTs in 2000 was $975 million, and by 2006, the amount increased to $2.17 billion. [13,14] In countries with a public healthcare system, it is difficult to quantify in dollars the cost of unnecessary imaging, but in Canada the rate of CT imaging has almost doubled since 2003, [15] suggesting that the cost of imaging has also drastically increased. This financial increase is also associated with trickle-down effects such as increased need for follow-up, further investigations of incidental findings, referrals to specialists, and even surgery.[10,16]

## Importance of assessing appropriateness

Given the potential patient harms and added health care costs of using diagnostic imaging, it is essential to understand if these tests are being used appropriately according to the current guidelines. This information allows healthcare providers to understand whether and to what degree patient safety and quality of care are compromised with the use of unnecessary tests. A recent systematic review of diagnostic imaging appropriateness for LBP found that approximately one third of imaging referrals were not appropriate; however, this review included imaging referrals from any healthcare provider for any imaging modality (including MRIs). [17] X-ray and CT pose the most direct harm to patients due to their radiation emissions; thus we intend to provide a focused estimate of appropriateness for these tests only. Additionally, since physicians in family practice or emergency department settings are the most common setting for imaging referrals for patients with LBP and follow the same guidelines for imaging ordering, we will focus our question to this provider population. This will also allow us to reduce any heterogeneity in our estimate due to potentially different ordering practices or guidelines amongst different providers.

## Aim

We aim to synthesize the evidence from all studies investigating the appropriateness of physician-made referrals for CTs and x-rays for LBP in primary and emergency care, which from here on we will refer to both as primary care. Our review adds to the literature by providing clinicians, implementation researchers and policy makers with an estimate of imaging appropriateness for CT imaging and x-ray imaging separately that is specific to physicians working in family practice and emergency department settings.

## Methods

This study was performed according to the PRISMA methodology.

## Search strategy

Four databases, PubMed, CINAHL, EMBASE and The Cochrane Database of Systematic Reviews, were searched for terms related to the PICO keywords of low back pain, guidelines, and adherence. The search string was developed with a research librarian. Databases were searched from inception to May 2018 (see S2 Appendix). Titles and abstracts from each

database search were imported to Endnote (version 10), and duplicates were removed before screening. Forward and backward citation tracking as well as reference lists of relevant systematic reviews and policy documents were done on all included papers in order to ensure our database search captured all applicable published research articles.

## Inclusion criteria

Studies were included if (i) the design was a retrospective or prospective review/audit of medical records, (ii) the data item was data on lumbar CT and x-ray images, (iii) the imaging referrals were made by a physician in either general practice or emergency department settings, (iv) the analysis compared the reason for imaging referral to a guideline source, and (v) the outcome was the proportion of appropriate or inappropriate referrals based on adherence to the guidelines. All LBP types were eligible for inclusion. We excluded studies that looked at appropriateness of imaging referred by other providers such as chiropractors, physiotherapists, or nurse practitioners. Only studies that reported individual or aggregate data from chart reviews for CT and x-ray imaging were included. If other tests or imaging modalities (e.g., MRI) were combined with x-rays or CTs, the study authors were contacted to confirm if x-ray and CT data could be reported separately, if not, the study would be excluded. Other study designs, such as self-reported surveys or simulated patient visits were excluded. Since there was potential for variation in imaging recommendations found in guidelines published prior to the year 2000 that could impact in the definition of appropriateness, we excluded all studies in which the data and guidelines were from 2000 and older.

Two reviewers (GL, AH) screened titles and abstracts and created a shortlist of full texts to be screened. Full texts were scrutinized by two reviewers (GL, AH) to assess eligibility against the inclusion/exclusion criteria. Any discrepancy was resolved upon discussion of the difference and consensus of the categorization for inclusion. Authors of studies that did not have a full text available (abstract or conference proceedings only) were contacted to determine if there was a published full-text. Authors of studies that did not report imaging modalities included were contacted to determine if MRI was included in the aggregate data.

## Data extraction

An electronic data collection form was developed to extract information from all included studies on study characteristics and outcome data. For each study the healthcare setting, LBP type, sample size, and outcome data were extracted. Outcomes included both the proportion of appropriate and inappropriate images. Additional outcome information extracted included: the guidelines source used for comparison, the definition used to assess appropriateness (or inappropriateness), the outcome denominator (if outcome reported the number of patients, images, visits), and measurement error (if reported).

## Quality of reporting and risk of bias assessment

Quality of reporting was assessed for each study according to the "Reporting of studies Conducted using Observational Routinely-collected health data" (RECORD) Statement checklist, which is an expansion of the "Strengthening the Reporting of Observational Studies in Epidemiology" STROBE Statement checklist.[18–21] Every included study was compared to the RECORD Statement's 35-item checklist to determine if the study reported pertinent information to fulfill the checklist.

No widely accepted tool exists for assessing Risk of Bias (RoB) for this type of observational study. Guidance was provided by a review authored by Sanderson et al. which provides a list of specific domains to be considered.[22] RoB for these observational, non-randomised studies

was determined by using items that related to the following 4 domains: Representativeness of patients, misclassification of patients, misclassification of outcome measurement, and inconsistent data. Overall study RoB was judged to be low if 4 out of the 4 domains were judged as low risk, moderate if 3 domains were considered low risk or high if two or less domain items were low risk.

## Data synthesis and analysis

Our main outcome was appropriateness of x-ray or CTs. For this review CT and x-ray appropriateness was broadly defined as suspicion of any of the red flag conditions (fracture, cauda equina, infection, malignancy). Since there is some variation in the guidelines about the exact criteria for appropriateness we anticipated some clinical heterogeneity in the definitions used by studies. Data were summarized separately for appropriateness of x-rays and appropriateness of CTs. We extracted estimates of the proportion of appropriate x-rays or CTs (and 95% confidence intervals) from each included study. In one case, the study only included an estimate of inappropriateness.[48] In this case the authors were contacted to confirm'that we could accurately use the inverse of their estimate as the proportion of appropriate x-rays. When studies did not provide CIs for their appropriate percentage, we calculated the 95% CI using the formula for calculating confidence intervals for a single proportion in Stata (v 15). Meta-analysis for a single proportion using a random effects model was completed on studies that were determined to be clinically homogenous.[23] The pooled proportion was calculated with Stata (v 15).

We applied the GRADE (Grading of Recommendations, Assessment, Development and Evaluation) approach to assess certainty of the estimates of appropriateness.[24] Certainty was downgraded based on 4 factors:

- Risk of Bias: Twenty-five percent or more of the participants were from studies rated as having a high RoB.

- Inconsistency in results: Determined by examining whether the estimates were similar in magnitude (overlapping confidence intervals).

- Indirectness of evidence: More than 50% of the participants were outside the target group (e.g., differences in populations, outcome measures, and interventions).

- Imprecision of evidence: Determined based on the width of the confidence interval (CI) associated with the proportion of appropriateness (+/- 3%) and the overall sample size (at least 2000 participants).

## Results

We identified a total of 919 publications from database searching (n = 918) and additional sources (n = 1), which was reduced to 696 studies after deduplication (Fig 1). We reviewed 185 full texts of which 22 were excluded for very specific reasons (see S2 Appendix).[25–46] Of the six final included studies,[47–52] one study was published in Spanish but was translated for analysis,[52] and two studies were abstracts only for which there was no full publication according to the authors of the abstracts.[47,48]

## Study characteristics

The studies were conducted in Finland, Ireland, Spain, & the United States (Table 1). In all studies, imaging referrals were made by physicians from a mixture of both primary care clinics or hospital settings. Sample sizes ranged from 30 to 3908. The duration of LBP in the different

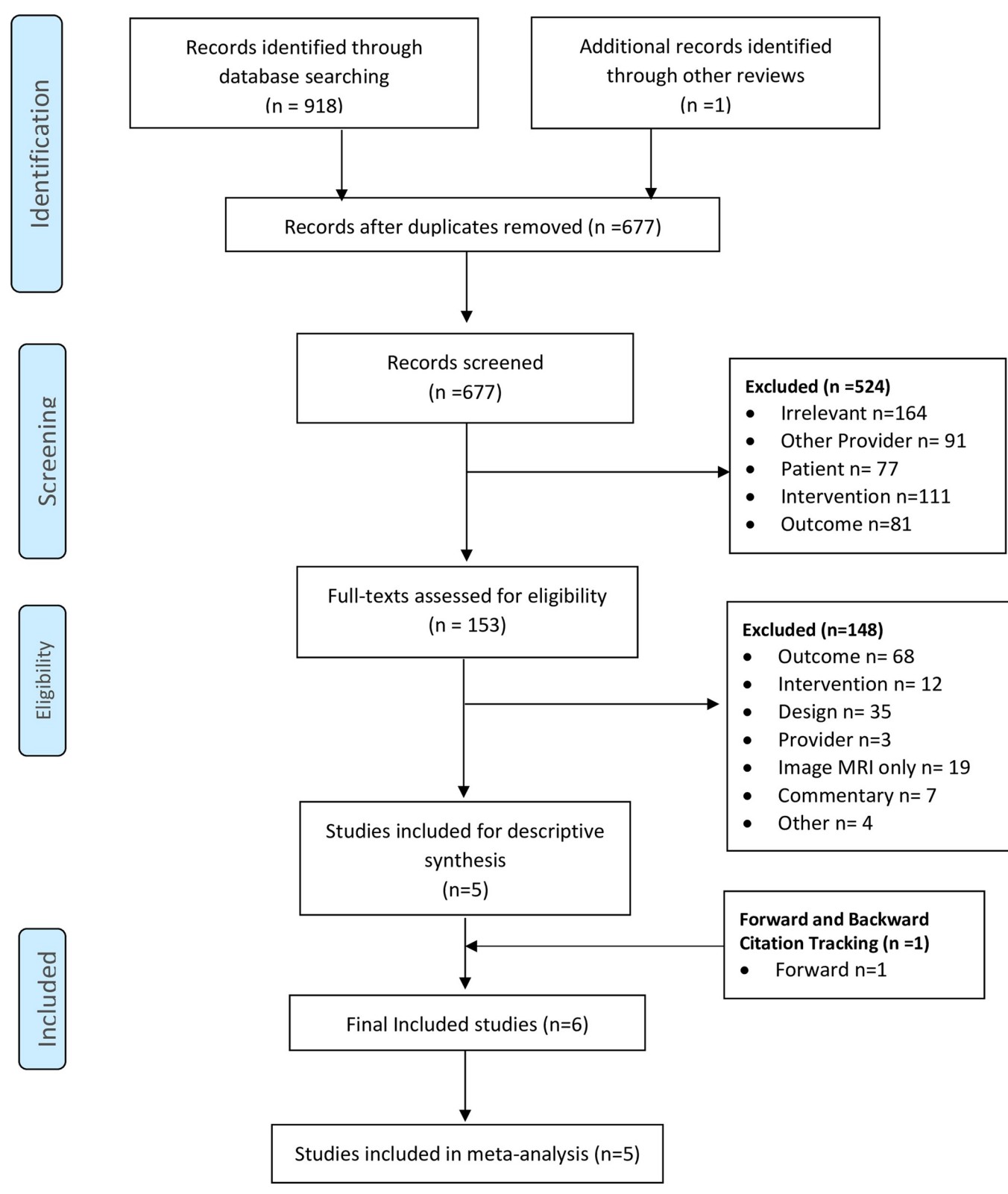

**Fig 1. PRISMA flow diagram of search strategy.**

**Table 1. Study characterised and reported outcomes of appropriateness organised by image type.**

| Study / Country | Setting[1] Patient age | Database / Data source | Definition of Appropriateness (Guideline Source) | Denominator (sample size)[2] | % Appropriate (95%CI) | Risk of Bias |
|---|---|---|---|---|---|---|
| *x-ray* | | | | | | |
| Baez 2011 USA | Mixed 18-40years | EMR/ Imaging referral[3] | Adherence to ACR, ACP and APS guidelines | Consecutive patients (18-40yrs) who received lumbar spine imaging (n = 100) | 34% (25, 43%) | High |
| Culleton 2013 Ireland | Mixed ≥65years | EMR/ Radiology findings | Adherence to RCR guidelines | All referrals for lumbar spine x-rays in patients >65yrs over a 5 month period (n = 414) | 18% (14, 22%) | High |
| Muntion-Alfaro 2006, Spain | Mixed NR | Medical Records/ Unclear | Adherence to red flag indicators listed in RCGP, AHCPR, and ICSI guidelines | Consecutive patients who presented at 1 GP clinic with low back pain who received a referral for an x-ray exam over a 1 year period. (n = 126) | 47% (43, 51%) | Moderate |
| Schlemmer 2015 USA | ED NR | Insurance Claims/ Imaging referral[3] | Adherence to red flag indicators, or >6-weeks of LBP as listed in the ACR and NCQA guidelines | All patients with a claim for a lumbar spine x-ray examination over a 1 year period. Note: this included only one x-ray claim per patient. (n = 3908) | 56% (55, 58%) | Low |
| Tahvonen 2016 Finland | Mixed NR | Medical Records/ Imaging referral Medical notes | Unclear (EC) | Consecutive patients (>16yrs) who received lumbar spine imaging referrals over a 6 month period (n = 50) | 32% (19, 45%) | High |
| *CTs* | | | | | | |
| Oikarinen 2009 Finland | Mixed < 35years | Medical Records Imaging referral[3] | Adherence to symptoms of fracture as listed in EC guidelines | Consecutive patients (<35yrs) who received a lumbar spine CT examination starting in January 2005 (n = 30) | 23% (8, 39%) | High |
| Schlemmer 2015 USA | ED NR | Insurance Claims Imaging referral[3] | Adherence to red flag indicators, or >6-weeks of LBP as listed in the ACR and NCQA Guidelines | All patients with a claim for a lumbar spine CT examination over a 1 year period. Note: this included only one CT claim per patient. (n = 648) | 56% (52, 60%) | Low |

[1] A mixed setting refers to studies that used a data source of imaging referrals in which the referring physician could be practicing in a family practice, in-hospital or emergency department setting.

[2] The total number of lumbar spine imaging/referrals reviewed.

[3] In addition to the referral, patient charts may have been accessed to determine patient information for determining appropriateness

NR: not reported.

EBG: Evidence Based Guidelines.

**Guideline Abbreviations**: NCQA: National Committee for Quality Assurance; RCGP: Royal College of General Practitioners; AHCPR: Agency for Health Care Policy and Research; ICSI: Institute for Clinical Systems Improvement; RCR: Royal College of Radiologists; ACR: American College of Radiologists; ACP: American College of Physicians; APS: American Pain Society; EC: European Commission

The type of low back pain (e.g. acute, chronic) was not specified in any of the studies.

Reporting quality using the RECORD checklist

studies was undefined. Five of 6 studies assessed appropriateness of x-rays; two of the six studies assessed appropriateness of CTs. The studies used a range of different guidelines to select the criteria for determining appropriateness. Of the six studies included, nine different guidelines were used; some studies were directed by more than one guideline source.

**Study design.** The included studies were all retrospective chart reviews/audits (see S2 Appendix), though not all used common terms to indicate that.[47] The majority of studies were a general chart audit/review done specifically to quantify appropriate imaging for LBP. However, one study's objective was to quantify appropriateness of CT imaging in young patients and included more than CT imaging of the lumbar spine (e.g., thoracic spine, head, etc.).[49]

**Setting.** All included studies were a general chart review of medical records and were conducted in a primary care provider setting and reported adequate information for the settings according to the RECORD checklist. The settings were identified as a hospital or health centre, with only one study mentioning data coming from the ED setting alone.[51]

| Author, year | Representativeness | Mis-classification Patient | Mis-classification Outcome | Inconsistent data | Final Judgement |
|---|---|---|---|---|---|
| Baez 2011 | ✔ | ✖ | ⚠ | ✔ | High |
| Culleton 2013 | ✔ | ✖ | ⚠ | ✔ | High |
| Muntion-Alfaro | ✔ | ✔ | ⚠ | ✔ | Moderate |
| Oikarinen 2009 | ✖ | ✖ | ✔ | ✔ | High |
| Schlemmer 2015 | ✔ | ✔ | ✔ | ✔ | Low |
| Tahvonen 2016 | ✔ | ✖ | ⚠ | ✔ | High |

**Fig 2. Risk of bias of included studies as determined by the representativeness of patients, risk of misclassification of patients, misclassification of the outcome of interest, and inconsistent data.**

**Participants and study size.** Participants were largely identified either by patient records, or records of images. Coding used to identify the included records was clearly described in only two studies.[51,52] These two studies were the only studies to justify their sample sizes.

**Data sources/variables.** Most studies took the information from the patients' hospital or clinic charts directly. If there was a specific database or computer program that was accessed, it was not communicated in the published paper. Electronic medical records were specified in three studies, but the applications were not identified by name.[48,51,52] One study utilized an insurance claims database.[51]

**Data access, cleaning, linkage, and supplementary information.** These reporting criteria were poorly or not at all discussed in the studies. If there was linkage involved it was not clarified and if the data cleaning occurred the details were not explained sufficiently. No study mentioned the level of database access researchers had. Only Schlemmer et al. provided supplementary data that was available for access online.[51]

**Risk of bias.** The four domains that were assessed for RoB were representativeness of patients, misclassification of patients, misclassification of outcome measurement, and inconsistency in data reporting (Fig 2). Four studies were judged to have a high risk of bias, one to have moderate RoB[52] and one to have low RoB.[51]

## Estimates of appropriateness

**X-rays.** We found five studies with 4,598 participants that reported the appropriateness of x-rays, with four studies that used the reason for referral to determine appropriateness (Table 1)[47,50–52] One study, by Culleton et al., used the radiology findings report interpreting the image to determine appropriateness.[48] It was excluded from the meta-analysis due to the heterogeneity of outcome assessment and data source. From the four studies with 4,184 participants, we found low quality evidence that 43% (95% CI: 30%, 56%) of x-rays were appropriate (Fig 3). The quality of evidence was downgraded for two reasons; inconsistency and indirectness (Table 2). The estimate was determined to be inconsistent based on non-overlapping confidence intervals of individual estimates across studies. As well, the estimate was downgraded due to indirectness as one of the studies was conducted solely in an ED setting while all others were in a mixed setting health centres with both general and ED physicians.

**CTs.** We found two studies with 678 participants that reported the appropriateness of CTs (Table 1). Both studies used the reason for referral to determine appropriateness but used

**Fig 3. Meta-analysis for proportion of appropriate x-rays and CT scans for low back pain.**

different criteria to define the outcome. Schlemmer et al.[51] defined appropriateness as any red flag condition or pain that has persisted greater than 6 weeks and Oikarinen et al.[49] restricted the definition to only situations of trauma. Using both studies, we found very low-quality evidence that 54% (95% CI: 51%, 58%) of CTs for LBP were appropriate (Fig 3). Similar to the outcome of x-ray appropriateness, the certainty of the estimate for CT appropriateness was downgraded due to inconsistency because of non-overlapping confidence intervals and indirectness because there were differences in the setting that would influence the outcome. Additionally, the estimate was downgraded due to imprecision, although the confidence intervals were somewhat narrow, the estimate is based on a sample size that is less than 2000 participants which challenges the certainty of the estimate (Table 2).

## Discussion

Few studies have been published reporting on the appropriateness of x-ray and CT scans ordered by primary care physicians (in general practice or emergency medicine) individually for patients with LBP. Among the studies we identified, most were conducted in European countries. No audit was conducted in countries such as Canada and Australia despite these countries having ongoing national campaigns to reduce unnecessary imaging for LBP (e.g., Choosing Wisely Canada, etc.).[7] From the available evidence, we found that only half of x-rays and CTs are being ordered according to guidelines. However, due to several factors

**Table 2. GRADE summary of findings for the outcome of appropriateness of x-ray and CT imaging for patients with low back pain.**

| Appropriateness of x-ray and CT imaging in patients with LBP ordered by primary and emergency care physicians | | | |
|---|---|---|---|
| Population: Patients with any type of low back pain<br>Setting: Emergency department, General Practice, Hospital<br>Comparison: Back pain guidelines for imaging, assumed to focus on red flag indicators | | | |
| Outcome | Effect | Number of participants in Studies | Certainty |
| Appropriateness of x-ray | 43% (30 to 56%) | n = 4,184; four studies | Low[2,4] ⨁⨁OO |
| Appropriateness of CTs | 54% (51 to 58%) | n = 678; two studies | Very low[2,3,4] ⨁OOO |

* GRADE Working Group grades of evidence.

**High quality:** Further research is very unlikely to change our confidence in the estimate of effect.

**Moderate quality:** Further research is likely to have an important impact on our confidence in the estimate of effect and may change the estimate.

**Low quality:** Further research is very likely to have an important impact on our confidence in the estimate of effect and is likely to change the estimate.

**Very low quality:** We are very uncertain about the estimate.

[1] Downgraded due to Risk of Bias

[2] Downgraded on Inconsistency

[3] Downgraded imprecision

[4] Downgraded on indirectness

related to inconsistency and indirectness, we have low certainty in this estimate. Our lack of certainty stems largely from the variation or lack of reporting how appropriateness had been defined in these studies. Moreover, the majority of the studies we identified were conducted with very small sample sizes (and were thus underpowered to provide reliable estimates) and were of low methodological and reporting quality. In order to advance the science in this area, better quality studies that are adequately powered and adhere to guidelines for conducting and reporting clinical audits using routinely collected data are required.

While another systematic review has investigated imaging appropriateness, it had heterogeneity by including multiple providers and included multiple imaging modality types, including MRI.[17] Our review adds to the current knowledge base in this area by answering a specific question regarding the appropriateness of radiation emitting x-ray and CT for patients with LBP in settings where patients typically seek care. Given that there have been several recent (past 5 years) international campaigns targeting physicians in general practice and emergency departments to reduce imaging, providing a robust assessment of the appropriateness specific to this recommendation is necessary to help clarify the issue and set targets for change.[7]

With respect to the estimate of imaging appropriateness, it is important to discuss that we found wide variation in the methods and reporting of the included studies. The six included studies cited 9 different guideline sources, which were not always internationally recognized. In addition, although the names and sometimes references of guidelines were mentioned as the source for determining appropriateness, it was not clear which criteria were used to define the outcome. For example, many guidelines recommended imaging only when red flags were present, and others provided additional criteria, which recommended imaging after a certain duration of LBP and non-response to treatment. It was unclear how these criteria were operationalized to code the reasons for referral as appropriate or not. This could lead to misclassification of the outcome or low reliability of the results. Better reporting of criteria for defining appropriateness and examples of operationalizing the coding protocol would improve our understanding of possible heterogeneity in the outcomes across studies.

Other sources of potential heterogeneity included the differences in inclusion criteria regarding patient population, the setting in which imaging referrals were made, and the medical record data sources. For example, two studies looked at patients that were under the age of 40, while one study looked only at patients older than 65 years. While most studies included a mixture of settings with referrals made from hospital-based or general practice-based physicians, one study focused solely on referrals made within an emergency department setting. Lastly, one study collected data from an insurance database, while two looked at EMR, and three did not describe the database other than to mention medical records. These potential sources of clinical heterogeneity may explain some of the inconsistency in the estimates across studies.

## Strengths

As with most systematic reviews and meta-analyses, we adhered to the PRISMA guidance for conducting and reporting systematic reviews and meta-analysis using observational data. [53,54] This included a) having two reviewers screen studies and extract data, b) providing an assessment of methodological quality and heterogeneity among the included studies, and c) forward and backward citation tracking to ensure all relevant studies were captured. We focused on an exact question of what the pooled proportion of radiation emitting imaging for patients with LBP in ED and primary care settings were appropriate which allowed us to understand how frequent these test orders are appropriate for these modalities that also cause harm to patients. Exclusion of older guidelines allows us to focus on recent studies that are

most applicable to the current guideline recommendations and current health care provider practice. Finally, we used the "RECORD checklist" to provide a robust assessment of the quality of reporting which allowed us to make sound recommendations for advancing the quality and replicability of the science in these types of study designs.

## Limitations

Despite its strengths, this study is limited in a few ways. First, due to resource constraints we chose to use a more specific search strategy meaning that it may not have been sufficiently sensitive to identify an exhaustive list of all potentially relevant studies. However, after consultation with a research librarian about this decision we included forward and backward citation tracking to enhance our specific search of electronic databases. While additional citation tracking did identify several potentially relevant studies all but one[51] were later excluded for various reasons (see S2 Appendix).

Other limitations of this systematic review involve the quality, risk of bias assessments, and heterogeneity of the included studies. Many of the studies were not described in sufficient detail to assess the quality for replicability. Since a tool does not already exist to help grade the studies that are reporting routinely collected health data, the domains for potential introduction of bias were selected based on expert opinion. This makes it difficult to compare to other systematic reviews. As mentioned, the clinical heterogeneity of the included studies with respect to the definition of appropriateness and differences in the inclusion criteria of patient ages also limits the certainty of our findings around the estimate of appropriateness, which we have reflected in our GRADE assessment.

## Future research

Based on this review's findings, we identified several areas for future research that would improve our knowledge about the appropriateness of LBP imaging. First, only 2 studies assessed the appropriateness of CT images for LBP that were ordered by physicians. One of these studies had a very small sample size and high risk of bias and the other was methodologically sound but was conducted in an ED setting. Future studies in other countries, using similar methods to Schlemmer et al. in both general practice and emergency settings, would be helpful to confirm appropriateness of CTs for LBP. This would involve adhering to the RECORD statement for improved reporting quality. Additionally, for both outcomes of x-rays and CTs, we found that the definition of appropriateness varied among studies and in many cases the definition was often unclear or too vague to allow meaningful interpretation or replication. Thus, as a first essential step, we recommend future research clearly report the definition of appropriateness they are using and the operationalization of the definition for coding purposes. Second, and possibly most important, this field of research would benefit from a standardized definition of appropriateness for x-rays and CTs. This could be based on a spectrum to reflect some variation in the guidelines, ranging from a very strict cut-off (e.g., appropriate if only trauma-indicated used in the Oikarinen et al. study) to more inclusive definitions (e.g., any red-flag indicated and/or having pain greater than 6 weeks as was used in Schlemmer et al).[49,51]

## Implications for practice

The results of this systematic review show that in several countries about half of the referrals for LBP imaging (x-rays and CTs) are not appropriate according to the guidelines. Due to the associated patient harms of x-ray and CTs scans including radiation exposure, high rates of incidental findings and risk of delayed recovery, non-adherence to the guidelines represents

low-value care for patients.[27] Hence, it is important to better understand why these referrals are made through future research.

## Conclusion

Recently there has been a push to reduce unnecessary and inappropriate imaging, not only to save costs, but also to provide better patient care.[10] This review provides an estimate of appropriateness for radiation emitting imaging for LBP, which indicates that only about half of imaging is appropriate according to recent guidelines. However, due to lack of published research, this estimate was not informed by data from many of the countries promoting the reduction of inappropriate imaging such as Canada, Australia and the UK. Moving forward, what we need is for more countries to undertake high quality studies with sufficiently large sample sizes using clear definitions of appropriateness.

## Supporting information

**S1 Appendix. Search strategies.**
(DOCX)

**S2 Appendix. Studies identified in search strategy (including forward and backward tracking) and the reason(s) they were excluded from descriptive synthesis and meta-analysis.**
(DOCX)

**S3 Appendix. RECORD and STROBE checklist items for included studies in descriptive synthesis.**
(DOCX)

## Acknowledgments

Thank you to Michelle Swab, a research librarian, who assisted with constructing the search strategy and applying it to different databases.

## Author Contributions

**Conceptualization:** Gabrielle S. Logan, Amanda Hall.

**Data curation:** Gabrielle S. Logan.

**Formal analysis:** Gabrielle S. Logan, Bethan Copsey.

**Funding acquisition:** Gabrielle S. Logan, Patrick Parfrey, Holly Etchegary.

**Investigation:** Gabrielle S. Logan.

**Methodology:** Gabrielle S. Logan, Amanda Hall.

**Project administration:** Gabrielle S. Logan, Amanda Hall.

**Resources:** Gabrielle S. Logan, Amanda Hall.

**Software:** Amanda Hall.

**Supervision:** Amanda Hall.

**Writing – original draft:** Gabrielle S. Logan.

**Writing – review & editing:** Gabrielle S. Logan, Andrea Pike, Bethan Copsey, Patrick Parfrey, Holly Etchegary, Amanda Hall.

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
