## [Decision Letter · Decision Letter 0]

28 Aug 2019

PONE-D-19-16551

What do we really know about the appropriateness of radiation emitting imaging for low back pain in primary care? A systematic review and meta-analysis of medical record reviews

PLOS ONE

Dear Mrs. Logan,

Thank you for submitting your manuscript to PLOS ONE. After careful consideration, we feel that it has merit but does not fully meet PLOS ONE’s publication criteria as it currently stands. Therefore, we invite you to submit a revised version of the manuscript that addresses the points raised during the review process.

The review question is important, but there are several aspects of the paper that need to be clarified. The three reviewers provided consistent recommendations for changes and I would strongly encourage that the authors address all the comments (particularly those made by reviewer #1). The introduction is quite long and I would suggest to cut some words. The authors should also carefully justify what their analyses add to the Jenkins et al (2018) review, particularly in light of all the sensitivity analyses they presented.

We would appreciate receiving your revised manuscript by Oct 12 2019 11:59PM. To enhance the reproducibility of your results, we recommend that if applicable you deposit your laboratory protocols in protocols.io, where a protocol can be assigned its own identifier (DOI) such that it can be cited independently in the future. For instructions see: http://journals.plos.org/plosone/s/submission-guidelines#loc-laboratory-protocols

We look forward to receiving your revised manuscript.

Kind regards,

Gustavo Machado, PhD

Academic Editor

PLOS ONE

Journal Requirements:

3. Please amend your manuscript to include your abstract after the title page.

Reviewers' comments:

Reviewer's Responses to Questions

**Comments to the Author**

1. Is the manuscript technically sound, and do the data support the conclusions?

Reviewer #1: Partly

Reviewer #2: Yes

Reviewer #3: Yes

2. Has the statistical analysis been performed appropriately and rigorously? 

Reviewer #1: Yes

Reviewer #2: Yes

Reviewer #3: N/A

3. Have the authors made all data underlying the findings in their manuscript fully available?

Reviewer #1: Yes

Reviewer #2: Yes

Reviewer #3: Yes

4. Is the manuscript presented in an intelligible fashion and written in standard English?

Reviewer #1: Yes

Reviewer #2: No

Reviewer #3: Yes

5. Review Comments to the Author

Reviewer #1: Thank you for asking me to review this interesting paper. The paper presented findings from a systematic review and meta-analysis specific to the appropriate use of X-ray and CT for low back pain. The paper reported generally sound methodology and is well written; however, I have a few concerns regarding the included papers and data analysis.

Major revisions:

1. Two of the included studies (Baez, 2011 and Culleton, 2013) are only referenced in abstract form. Were full studies published to enable complete data extraction and analysis of risk of bias, or were the authors contacted for further details? More information regarding this should be included, or the reference should be changed to that of the published full text. In addition, one study (Muntion-Alfaro, 2006) was published in Spanish with only the abstract available in English - was the full text translated?

2. More detail regarding the numerator and denominator extracted from the studies to calculate the proportion of appropriate imaging is required. For example was it the proportion of: number of appropriate image referrals/total number of image referrals OR was it the proportion of: number of patients referred for imaging/number of patients determined as appropriate for imaging? Both give different measures of appropriateness but with different denominators they are not directly comparable. Currently it is unclear which measure has been used, or whether both have been used which will impact on the suitability of meta-analysis. This should be made clear in Table 1.

3. The suitability of the meta-analyses I think needs to be further considered. It is unclear in the methods what factors were considered when assessing for clinical homogeneity and this should be further described. The CT meta-analysis only includes 2 studies, of which one is weighted 94% so I am questioning the suitability of pooling these, especially as one of the studies had more limited determination of appropriateness compared to the other (ie. trauma indications only). For the X-ray meta-analysis it is unclear if the outcome measures are comparable (see point 2 above).

Minor revisions:

Abstract-

1. The conclusion is not clearly articulated - try to make the main finding of the review and possible implications more clear.

Introduction -

1. The introduction is quite long and I feel the section of harms of over testing could be summarised more succinctly. Given the aim of doing a review specific to X-ray and CT as opposed to including MRI, some more explanation of why you chose to do this could be useful. In particular you state that X-ray and CT provide the most direct harm, but with no references to support this. Although MRI doesn't have ionising radiation it could arguably reveal more incidental findings and have more costs associated. The use of CT/MRI has also been increasing over time (Downie A, Hancock M, Jenkins H, et al How common is imaging for low back pain in primary and emergency care? Systematic review and meta-analysis of over 4 million imaging requests across 21 years British Journal of Sports Medicine Published Online First: 13 February 2019. doi: 10.1136/bjsports-2018-100087).

2. Page 2 under 'Harms of over-testing' references are required for the first 2 statements.

3. Page 3 top paragraph - I would also consider the potential increased risk of carcinogenic changes in children

4. Page 4 top[ paragraph, line 3. Swap is and also: This financial increase is also associated with...

Methods -

1. Was the review registered in Prospero, if so please provide details.

2. Page 5 Under inclusion criteria the numbers (i), (ii) etc. for the five points are repetitive and not sequential in a list: (ii) and (iii) are both listed twice - please modify

3. Page 6 Excluded studies prior to 2000 due to guidelines - but wouldn't this more depend on which guidelines were used rather than the year of data collection (later studies may have used older guidelines) - perhaps consider exclusion on the type of guideline.

4. Page 6 Under data extraction, last sentence: remove the 'was extracted' from the end of the sentence and add 'extracted' to the beginning of the sentence - 'Additional outcome information extracted included...'

Discussion -

1. Page 15 paragraph 2. I would disagree with the statement: 'Prior to our review, it was difficult to say anything regarding the appropriateness of imaging for LBP according to the guidelines'. I am an author on the 2018 review into this topic that is then discussed in the same paragraph. This review also looked at imaging appropriateness and conclusions can be made from the data presented in the review (indeed, you referenced one of these conclusions in the introduction). Although the 2018 review did have heterogeneity of included studies as mentioned, the data analysis in the review accounted for this by performing different meta-analyses with respect to the guidelines used to assess appropriateness and the outcome measure used, and by performing sensitivity analysis to account for clinical setting, type of imaging and year of publication. In this paragraph it would be better to see a comparison of the results of your study to that of the previous review with a discussion of possible reasons for similarities/differences, which would include the more specific inclusion criteria of the current review.

2. Page 15 paragraph 2. You state that there have been several campaigns to reduce X-ray and CT use but only provide one reference specific to Canada. I am also not certain whether such campaigns are specific only to Xray and CT in general, or whether they often include all imaging which would include MRI.

3. Page 15 paragraph 3 to Page 16 paragraph 2 - this information may be better moved to under limitations.

4. Page 16 Strengths: Most of the strengths you have listed are fairly standard practice for SLRs. Are there any particular strengths that you feel are more unique to your review - ie. in the way you analysed the data, the question you asked etc.

5. Page 16-17 Limitations: If the meta-analyses are left as is (after considering the points made above) then I feel there should be more discussion of the potential limitations of these.

Conclusion -

1. Statement again 'Before this review, it was difficult to say anything regarding how appropriate imaging for LBP is according to the guidelines'. I would again disagree with this statement as discussed above. I would remove this and re-phrase the conclusion accordingly

Reviewer #2: I thank the authors for the opportunity to review this manuscript. The authors aimed to investigate the proportion of XR and CT imaging requests for low back pain that were appropriate. This is an extremely important question. Reducing the inappropriate use of imaging is a priority for numerous healthcare organisations and initiatives that aim to reduce low-value care (e.g. Choosing Wisely). However, before resources are spent on strategies to reduce imaging, it is important to understand the size of this problem.

Although the review question is important, I don't think the rationale is strong enough for why this review is sufficiently different from the review by Jenkins et al (2018). The authors should carefully justify what their analyses add to the Jenkins et al (2018) review, particularly in light of all the sensitivity analyses that are presented in Table S5 (https://www.sciencedirect.com/science/article/pii/S1529943018302031?via%3Dihub#ec0015).

There are also numerous issues with grammar that need to be addressed. For example, the following phrases in the Abstract need to be revised:

- 'pooled proportion of appropriateness of CT and XR imaging for low back pain' should be 'pooled proportion of CT and XR imaging for low back pain that were considered appropriate

- 'Four studies reported XR appropriateness, one study reported CT appropriateness should be ' Four studies reported on the appropriateness of XR imaging, one on the appropriateness of CT, ...'

- the abstract conclusion is similar

- The authors should carefully scan the manuscript for similar examples and correct them

Abstract

- I am unclear what the RECORD checklist is from just reading the abstract. Is it possible to provide a brief explanation in the abstract methods?

- I think the abstract conclusion could better reflect the results. For example, 'There is low to very-low quality evidence that only half of XRs and CTs ordered for LBP are appropriate

- I would also add the need for future research to properly examine 'appropriateness' given the low quality of the evidence

Introduction

- remove the abbreviation for diagnostic imaging (DI) as it is not a commonly used phrase

-Page 3, 1st paragraph: the authors need to acknowledge that CT exposes patients to substantially more potentially harmful radiation than XRs

-Page 3, 2nd paragraph: can the authors also provide data for younger age groups?

-Page 3, 2nd paragraph: the authors could also mention that incidental findings can lead to surgery

-Page 4, 2nd paragraph: the authors should elaborate on why CT and XR post more direct harms to patients when compared with MRI. I’m not really sure why MRI imaging was excluded from this review.

Method

-Page 8: For the GRADE criteria 'indirectness of evidence', could the authors provide an example of participants being outside the target group?

Results

-Page 13, 1st paragraph: I think it is a big assumption that all people presenting to ED with LBP are doing so because of trauma. Do the authors have a reference to support this?

- Appendix 3 is difficult to interpret because there is no reference to what each item means. I suggest including a description of the items directly under the table.

Table 1

-in the column labelled 'definition of appropriateness', 'no red flags' and 'red flag indicators' appear to be contradictory. Wouldn't the presence of red flags be an indication for appropriateness?

-for 'Culleton 2013', there seems to be an 'NR' value included by mistake

- please add the setting for each study in this table (i.e. primary care or emergency)

Table 2

- the first row in table 2 mentions 'primary care physicians' but my understanding is that studies from ED were also included in this review. My understanding is that ED physicians are not primary care physicians. Could the authors please clarify this and ensure the terminology used throughout the manuscript is consistent in regards to this issue

Discussion

-Page 14, 1st paragraph: the authors could also make reference to the Choosing Wisely campaign in Australia

-Page 16, 3rd paragraph: please remove the use of a random effects meta-analysis as a strength of this review

Reviewer #3: Thank you for asking me to review this manuscript. This study is a systematic review and meta-analysis of appropriateness of radiation emitting imaging for low back pain.

The manuscript is well written. Please see below some minor comments/suggestions for improvement:

1. Introduction reads well.

2. It is unclear whether the protocol was registered/study followed a registered protocol.

3. Authors have mentioned that they searched Pubmed, CINAHL, and Embase in the abstract but mentioned four databases in the manuscript. Suggest adding the fourth database- The Cochrane Database of Systematic Reviews in the abstract as well.

4. Page 7 – it’s not really an ‘effect size,’ it’s a proportion or pooled proportion. Suggest changing these terms throughout e.g. in Table 2.

5. No data from low-middle income countries – discussion point

6. Authors have lumped proportions with different denominators (% of images vs % patients presenting for care). Does it make sense to do this? Perhaps pooling the proportions with the same denominator would be better. Probably ok to lump scan types in together

7. Are the numbers in Table 2 the number of patients presenting for care, or the number of patients who were referred for imaging? Please make this clear in the manuscript.

8. Details on the number of studies assessing appropriateness of x-rays and CTs do not match in the abstract and manuscript. I’d suggest using consistent language to avoid confusion. For eg, abstract says “Four studies reported x-rays appropriateness, one study reported CT appropriateness, and one study reported on both imaging modalities.” Manuscript says “Five of 6 studies assessed appropriateness of x-rays; two of the six studies assessed appropriateness of CTs”

9. In study methods the authors do not mention whether the study followed PRISMA guidance. It was only mentioned in the Strengths section. Please consider adding it in the Methods section as well.

10. In ‘Estimates of Appropriateness’ section, when describing x-Rays, I suggest adding

number of participants (similar to what you have done in ‘CTs’) to make it consistent.

Eg, we found five studies with 5010 participants that reported the appropriateness of x-rays.

11. Title of the study is “….appropriateness of imaging for back pain in primary care” but includes studies in emergency department and hospital settings.

12. From Table 1“A mixed setting refers to studies that used a data source of imaging referrals in which the referring physician could be practicing in a family practice, in-hospital or emergency department setting” Some clarity is needed on how the authors have defined primary care. In some healthcare systems hospital-based care is not considered primary care.

13. Suggest using word ‘imaging’ instead of ‘images’ in inclusion criteria, second point, page 5.

14. In ‘Data Access, Cleaning, Linkage, and Supplementary Information’ section, page 12, please add ‘for’ in the sentence ‘No study mentioned the level of database access researchers’.

15. Consider rewording the sentence- Of the six studies, nine different guidelines were used (in study characteristics, page 9).

6. PLOS authors have the option to publish the peer review history of their article (what does this mean?). If published, this will include your full peer review and any attached files.

Reviewer #1: No

Reviewer #2: No

Reviewer #3: Yes: Ms Sweekriti Sharma

Dr Adrian Traeger

---

## [Author Response · Author response to Decision Letter 0]

17 Oct 2019

Reviewer #1: Thank you for asking me to review this interesting paper. The paper presented findings from a systematic review and meta-analysis specific to the appropriate use of X-ray and CT for low back pain. The paper reported generally sound methodology and is well written; however, I have a few concerns regarding the included papers and data analysis.

Major revisions:

1. Two of the included studies (Baez, 2011 and Culleton, 2013) are only referenced in abstract form. Were full studies published to enable complete data extraction and analysis of risk of bias, or were the authors contacted for further details? More information regarding this should be included, or the reference should be changed to that of the published full text. In addition, one study (Muntion-Alfaro, 2006) was published in Spanish with only the abstract available in English - was the full text translated?

RESPONSE: Thank you for pointing this out for clarification. We did contact Baez 2011 and Culleton 2013 (and mentioned this in the Methods section on page 6), but neither author had published full manuscripts. The full article by Muntion-Alfaro 2006 was translated from Spanish into English. We have clarified this process in the manuscript (Pg 7 & 8).

2. More detail regarding the numerator and denominator extracted from the studies to calculate the proportion of appropriate imaging is required. For example was it the proportion of: number of appropriate image referrals/total number of image referrals OR was it the proportion of: number of patients referred for imaging/number of patients determined as appropriate for imaging? Both give different measures of appropriateness but with different denominators they are not directly comparable. Currently it is unclear which measure has been used, or whether both have been used which will impact on the suitability of meta-analysis. This should be made clear in Table 1.

RESPONSE: Thank you for your comment. We have added in the specific definitions to clarify the numerator and denominator used in each study in Table 1 on pg 10. Importantly, we want to be clear that in the calculations of appropriateness for each of the included studies we are confident that the numerator included the number of referrals that met the criteria for appropriateness and the denominator included the total number of images during the study period. In all cases the denominator included only one image per patient. Therefore, the denominators for the included studies are considered clinically homogenous

3. The suitability of the meta-analyses I think needs to be further considered. It is unclear in the methods what factors were considered when assessing for clinical homogeneity and this should be further described. The CT meta-analysis only includes 2 studies, of which one is weighted 94% so I am questioning the suitability of pooling these, especially as one of the studies had more limited determination of appropriateness compared to the other (ie. trauma indications only). For the X-ray meta-analysis it is unclear if the outcome measures are comparable (see point 2 above).

RESPONSE: Thank you for this comment and raising two important points regarding the rationale for choosing to include a meta-analysis in this review. On the first point regarding the CT scans and clinical homogeneity, we anticipated a level of clinical heterogeneity (particularly in how the outcome of appropriateness would be defined). For this review, appropriateness was defined as suspicion of any of the red flag conditions (fracture, cauda equina, infection, malignancy). 

Since our question was about appropriateness in general we made the choice to pool these estimates to answer our broad question. We have added this additional information to the data synthesis section on page 7. We decided to downgrade the quality of the evidence based on indirectness in cases where a large portion of the study data came from studies that focused on a sub-group of participants that may influence the estimate. This can be noted in our GRADE assessment. I hope this answers your questions.

Minor revisions:

Abstract-

1. The conclusion is not clearly articulated - try to make the main finding of the review and possible implications more clear.

RESPONSE: Thank you. We have reworded the conclusion statement in the abstract.

Introduction -

1. The introduction is quite long and I feel the section of harms of over testing could be summarised more succinctly. Given the aim of doing a review specific to X-ray and CT as opposed to including MRI, some more explanation of why you chose to do this could be useful. In particular you state that X-ray and CT provide the most direct harm, but with no references to support this. Although MRI doesn't have ionising radiation it could arguably reveal more incidental findings and have more costs associated. The use of CT/MRI has also been increasing over time (Downie A, Hancock M, Jenkins H, et al How common is imaging for low back pain in primary and emergency care? Systematic review and meta-analysis of over 4 million imaging requests across 21 years British Journal of Sports Medicine Published Online First: 13 February 2019. doi: 10.1136/bjsports-2018-100087).

RESPONSE: Thank you for this comment. However, we did not state that CT and x-ray expose patients to more direct harms than MRI. We simply mentioned that there are direct harms to the patient from these imaging types. We have edited the introduction to be more concise and have tried to clarify this misunderstanding.

2. Page 2 under 'Harms of over-testing' references are required for the first 2 statements.

RESPONSE: Thank you. As you have recommended decreasing the introduction, and since these sentences were not necessary, we have deleted them to be concise.

3. Page 3 top paragraph - I would also consider the potential increased risk of carcinogenic changes in children

RESPONSE: Thank you for this comment. We have added a minor comment on paediatric populations. We have refrained from further elaboration, as this is not the population of focus for this systematic review.

4. Page 4 top [ paragraph, line 3. Swap is and also: This financial increase is also associated with...

RESPONSE: Thank you for catching this grammar error. We have adjusted it.

Methods -

1. Was the review registered in Prospero, if so please provide details.

RESPONSE: This review was not registered in Prospero due to time restrictions on the project. 

2. Page 5 Under inclusion criteria the numbers (i), (ii) etc. for the five points are repetitive and not sequential in a list: (ii) and (iii) are both listed twice - please modify

RESPONSE: Thank you for catching this. We have corrected this mistake.

3. Page 6 Excluded studies prior to 2000 due to guidelines - but wouldn't this more depend on which guidelines were used rather than the year of data collection (later studies may have used older guidelines) - perhaps consider exclusion on the type of guideline.

RESPONSE: This is in fact what we were trying to communicate. We apologise for the lack on clarity and have reworded it (pg 6). 

4. Page 6 Under data extraction, last sentence: remove the 'was extracted' from the end of the sentence and add 'extracted' to the beginning of the sentence - 'Additional outcome information extracted included...'

RESPONSE: We have made this edit. Thank you.

Discussion -

1. Page 15 paragraph 2. I would disagree with the statement: 'Prior to our review, it was difficult to say anything regarding the appropriateness of imaging for LBP according to the guidelines'. I am an author on the 2018 review into this topic that is then discussed in the same paragraph. This review also looked at imaging appropriateness and conclusions can be made from the data presented in the review (indeed, you referenced one of these conclusions in the introduction). Although the 2018 review did have heterogeneity of included studies as mentioned, the data analysis in the review accounted for this by performing different meta-analyses with respect to the guidelines used to assess appropriateness and the outcome measure used, and by performing sensitivity analysis to account for clinical setting, type of imaging and year of publication. In this paragraph it would be better to see a comparison of the results of your study to that of the previous review with a discussion of possible reasons for similarities/differences, which would include the more specific inclusion criteria of the current review.

RESPONSE: Thank you for your comment. While we thought that this statement was broad enough, we understand your point of view and have removed the sentence.

2. Page 15 paragraph 2. You state that there have been several campaigns to reduce X-ray and CT use but only provide one reference specific to Canada. I am also not certain whether such campaigns are specific only to Xray and CT in general, or whether they often include all imaging which would include MRI.

RESPONSE: We only cited one Choosing Wisely Campaign here, but are aware that there are others in different jurisdictions. We were hoping to minimise our word count by only referring to one Choosing Wisely Campaign. However, we have edited the statement to be more accurate with Choosing Wisely recommendations. 

3. Page 15 paragraph 3 to Page 16 paragraph 2 - this information may be better moved to under limitations.

RESPONSE: Thank you for your suggestion. We think the section on heterogeneity of appropriateness definitions is an important point for the discussion for systematic reviews beyond its limitations. We have left this paragraph as is but we have also referred to it in the limitations as you have thoughtfully suggested. 

4. Page 16 Strengths: Most of the strengths you have listed are fairly standard practice for SLRs. Are there any particular strengths that you feel are more unique to your review – ie. In the way you analysed the data, the question you asked etc.

RESPONSE: Thank you for your comment. We have added some adjustments to our strengths section.

5. Page 16-17 Limitations: If the meta-analyses are left as is (after considering the points made above) then I feel there should be more discussion of the potential limitations of these.

RESPONSE: Thank you for this suggestion. We hope our previous responses have clarified our rationale for pooling the data and that our discussion of the clinical heterogeneity is satisfactory for the reviewer. We have also edited the limitations section to reflect how these decisions may impact the certainty of the estimates presented.

Conclusion -

1. Statement again ‘Before this review, it was difficult to say anything regarding how appropriate imaging for LBP is according to the guidelines’. I would again disagree with this statement as discussed above. I would remove this and re-phrase the conclusion accordingly

RESPONSE: We have taken your suggestion and removed the sentence. Thank you for your comment.

Reviewer #2: I thank the authors for the opportunity to review this manuscript. The authors aimed to investigate the proportion of XR and CT imaging requests for low back pain that were appropriate. This is an extremely important question. Reducing the inappropriate use of imaging is a priority for numerous healthcare organisations and initiatives that aim to reduce low-value care (e.g. Choosing Wisely). However, before resources are spent on strategies to reduce imaging, it is important to understand the size of this problem.

Although the review question is important, I don't think the rationale is strong enough for why this review is sufficiently different from the review by Jenkins et al (2018). The authors should carefully justify what their analyses add to the Jenkins et al (2018) review, particularly in light of all the sensitivity analyses that are presented in Table S5 (https://www.sciencedirect.com/science/article/pii/S1529943018302031?via%3Dihub#ec0015).

There are also numerous issues with grammar that need to be addressed. For example, the following phrases in the Abstract need to be revised:

- 'pooled proportion of appropriateness of CT and XR imaging for low back pain' should be 'pooled proportion of CT and XR imaging for low back pain that were considered appropriate

- 'Four studies reported XR appropriateness, one study reported CT appropriateness should be ' Four studies reported on the appropriateness of XR imaging, one on the appropriateness of CT, ...'

- the abstract conclusion is similar

- The authors should carefully scan the manuscript for similar examples and correct them

RESPONSE TO ABOVE FOUR POINTS: Thank you and apologies for these errors. We have made grammar edits as we have noticed them.

Abstract

- I am unclear what the RECORD checklist is from just reading the abstract. Is it possible to provide a brief explanation in the abstract methods?

- I think the abstract conclusion could better reflect the results. For example, 'There is low to very-low quality evidence that only half of XRs and CTs ordered for LBP are appropriate

- I would also add the need for future research to properly examine 'appropriateness' given the low quality of the evidence

RESPONSE: Thank you. Edits have been made.

Introduction

- remove the abbreviation for diagnostic imaging (DI) as it is not a commonly used phrase

-Page 3, 1st paragraph: the authors need to acknowledge that CT exposes patients to substantially more potentially harmful radiation than XRs

-Page 3, 2nd paragraph: can the authors also provide data for younger age groups?

-Page 3, 2nd paragraph: the authors could also mention that incidental findings can lead to surgery

-Page 4, 2nd paragraph: the authors should elaborate on why CT and XR post more direct harms to patients when compared with MRI. I’m not really sure why MRI imaging was excluded from this review.

RESPONSE TO ABOVE 5 POINTS: Thank you. We have removed the DI abbreviation and clarified that CT emits more radiation than XRs. We have briefly mentioned paediatric populations and the risk procedures from imaging. We have chosen not to expand on paediatric populations because that is beyond the scope of this paper and not the population of interest. For the sake of length we have also not expanded on surgery risks. Finally, we have clarified why CT and x-ray are the primary focus of this manuscript. Thank you for all of these suggestions and we hope we have addressed them adequately.

Method

-Page 8: For the GRADE criteria 'indirectness of evidence', could the authors provide an example of participants being outside the target group?

RESPONSE: We have added examples to the indirectness of evidence bullet point “(e.g., differences in populations, outcome measures, and interventions)”. Thank you for this suggestion.

Results

-Page 13, 1st paragraph: I think it is a big assumption that all people presenting to ED with LBP are doing so because of trauma. Do the authors have a reference to support this?

RESPONSE: Thank you for your insight and we agree that this is an assumption. We have removed the sentence, as it was not necessary for the point being made.

- Appendix 3 is difficult to interpret because there is no reference to what each item means. I suggest including a description of the items directly under the table.

RESPONSE: This appendix is based off of a large and thorough checklist created for reporting reliable information for observational studies. Adding in the description of each item in the checklist would significantly add to the size of this table, thus we have referenced the original table to ensure that this is easy to find.

Table 1

-in the column labelled 'definition of appropriateness', 'no red flags' and 'red flag indicators' appear to be contradictory. Wouldn't the presence of red flags be an indication for appropriateness?

-for 'Culleton 2013', there seems to be an 'NR' value included by mistake

- please add the setting for each study in this table (i.e. primary care or emergency)

RESPONSE: Thank you for catching these mistakes in our table 1. We have corrected the definition of appropriateness for the Muntion-Alfaro study, and removed the unnecessary NR in the Culleton row. The settings for each study is found in Table 1 in the second column.

Table 2

- the first row in table 2 mentions 'primary care physicians' but my understanding is that studies from ED were also included in this review. My understanding is that ED physicians are not primary care physicians. Could the authors please clarify this and ensure the terminology used throughout the manuscript is consistent in regard to this issue

RESPONSE: Thank you for this insight. While we understand that there is some debate regarding whether primary care is inclusive of the emergency department, in our context it is often used to refer to both general and emergency physicians, as they are the first point of contact into the healthcare system. We have edited our introduction to address this issue briefly and continue to discuss both settings as primary care. We hope this explanation is satisfactory.

Discussion

-Page 14, 1st paragraph: the authors could also make reference to the Choosing Wisely campaign in Australia

-Page 16, 3rd paragraph: please remove the use of a random effects meta-analysis as a strength of this review

RESPONSE: Thank you. We have removed the random effects sentence, and we did not make reference to other jurisdictions for the sake of word count. 

Reviewer #3: Thank you for asking me to review this manuscript. This study is a systematic review and meta-analysis of appropriateness of radiation emitting imaging for low back pain.

The manuscript is well written. Please see below some minor comments/suggestions for improvement:

1. Introduction reads well.

RESPONSE: Thank you for your comment.

2. It is unclear whether the protocol was registered/study followed a registered protocol.

RESPONSE: We appreciate the suggestion. However, we did not register a protocol due to time constraints.

3. Authors have mentioned that they searched Pubmed, CINAHL, and Embase in the abstract but mentioned four databases in the manuscript. Suggest adding the fourth database- The Cochrane Database of Systematic Reviews in the abstract as well.

RESPONSE: Thank you for your suggestion. We have added it to the abstract.

4. Page 7 – it’s not really an ‘effect size,’ it’s a proportion or pooled proportion. Suggest changing these terms throughout e.g. in Table 2.

RESPONSE: Thank you for your suggestion. We have made this change. 

5. No data from low-middle income countries – discussion point

RESPONSE: Thank you for this observation. Due to our already extended discussion we have chosen not to discuss this for the sake of discussion length.

6. Authors have lumped proportions with different denominators (% of images vs % patients presenting for care). Does it make sense to do this? Perhaps pooling the proportions with the same denominator would be better. Probably ok to lump scan types in together

RESPONSE: Please see the response to Comment 3 from Reviewer 1, found on page 2. We hope that response adequately addresses the concerns from this reviewer and have revised Table 1 to be more clear. Thank you for your comments.

7. Are the numbers in Table 2 the number of patients presenting for care, or the number of patients who were referred for imaging? Please make this clear in the manuscript.

RESPONSE: We have clarified this point. Thank you.

8. Details on the number of studies assessing appropriateness of x-rays and CTs do not match in the abstract and manuscript. I’d suggest using consistent language to avoid confusion. For eg, abstract says “Four studies reported x-rays appropriateness, one study reported CT appropriateness, and one study reported on both imaging modalities.” Manuscript says “Five of 6 studies assessed appropriateness of x-rays; two of the six studies assessed appropriateness of CTs”

RESPONSE: Thank you. We agree that the wording is confusing, and have made changes to be clearer.

9. In study methods the authors do not mention whether the study followed PRISMA guidance. It was only mentioned in the Strengths section. Please consider adding it in the Methods section as well.

RESPONSE: This has been mentioned in the Methods section. Thank you for your suggestion.

10. In ‘Estimates of Appropriateness’ section, when describing x-Rays, I suggest adding number of participants (similar to what you have done in ‘CTs’) to make it consistent.

Eg, we found five studies with 5010 participants that reported the appropriateness of x-rays.

RESPONSE: We have made this change.

11. Title of the study is “….appropriateness of imaging for back pain in primary care” but includes studies in emergency department and hospital settings.

RESPONSE: This comment has been made by a previous reviewer and we have addressed it above.

12. From Table 1“A mixed setting refers to studies that used a data source of imaging referrals in which the referring physician could be practicing in a family practice, in-hospital or emergency department setting” Some clarity is needed on how the authors have defined primary care. In some healthcare systems hospital-based care is not considered primary care.

RESPONSE: This is a similar comment to another reviewer and we have addressed it above (pg 7 of this response letter). Thank you for your comment!

13. Suggest using word ‘imaging’ instead of ‘images’ in inclusion criteria, second point, page 5.

RESPONSE: We have made this change. Thank you.

14. In ‘Data Access, Cleaning, Linkage, and Supplementary Information’ section, page 12, please add ‘for’ in the sentence ‘No study mentioned the level of database access researchers’.

RESPONSE: There is in fact a word missing from that sentence and we have corrected it. Thank you for catching our mistake.

15. Consider rewording the sentence- Of the six studies, nine different guidelines were used (in study characteristics, page 9).

RESPONSE: We have made this change. Thank you.

---

## [Editor Report · Decision Letter 1]

5 Nov 2019

What do we really know about the appropriateness of radiation emitting imaging for low back pain in primary and emergency care? A systematic review and meta-analysis of medical record reviews

PONE-D-19-16551R1

Dear Gabrielle,

We are pleased to inform you that your manuscript has been judged scientifically suitable for publication and will be formally accepted for publication once it complies with all outstanding technical requirements.

With kind regards,

Gustavo Machado, PhD

Academic Editor

PLOS ONE
---

## [Editor Report · Acceptance letter]

12 Nov 2019

PONE-D-19-16551R1 

What do we really know about the appropriateness of radiation emitting imaging for low back pain in primary and emergency care? A systematic review and meta-analysis of medical record reviews 

Dear Dr. Logan:

I am pleased to inform you that your manuscript has been deemed suitable for publication in PLOS ONE. Congratulations! Your manuscript is now with our production department. 

With kind regards,

on behalf of

Dr. Gustavo de Carvalho Machado 

Academic Editor

PLOS ONE